# Performance of DBD Actuator Models under Various Operating Parameters and Modifications to Improve Them

Raul Alberto Bernal-Orozco [1], Ignacio Carvajal-Mariscal [1] and Oliver Marcel Huerta-Chavez [2,*]

1 Instituto Politecnico Nacional, ESIME-UPALM, Mexico City 07738, Mexico
2 Instituto Politecnico Nacional, SEPI-ESIME Ticóman, Mexico City 07340, Mexico
* Correspondence: ohuertac@ipn.mx

**Abstract:** Simulation is a valuable tool for the study of DBD actuators, therefore accurate, computationally efficient, and robust numerical models are required. The performance of three DBD actuator models was studied: the phenomenological Shyy and Suzen models, and the empirical Dörr and Kloker model. The first objective of this work is to determine the ability of these models to reproduce the force and induced flow by comparing the numerical results with experimental reference data reported in the literature. As a second objective, modifications have been proposed to improve these models. Several simulations were performed in OpenFOAM with different geometrical parameters, voltages, and frequencies. Discrepancies and limitations of the models were identified. The modified Dörr and Kloker model allows more consistent use of this model by considering a factor that relates it to voltage and frequency. Shyy's modified model reduces the overestimation of force and velocity. Suzen's modified model is the one that fits the reference data better, so its use is suggested over the other models. The proposed modifications are easy to implement and allow significant improvements in the capacity of the models to reproduce the effects of a DBD actuator.

**Keywords:** plasma actuators; dielectric barrier discharge plasma actuator; flow control; wall jet





## 1. Introduction

The need to improve the performance of aircraft and wind turbines has led to the study and development of active flow control methods. Among these techniques, plasma actuators, in particular surface dielectric barrier discharge (SDBD) or DBD actuators, have gained interest because of their fast response, lack of moving parts and low weight.

Several applications of DBD actuators have been explored, the most studied being drag reduction [1,2], increased lift [3–6], boundary layer control [7–9], and control of separation bubbles [10,11]. Recently, these devices have been studied for its use as anti-icing systems [12–15], for noise reduction [16,17] and film cooling [18,19]. Similarly, their use has been explored in various mechanisms, such as wind turbines (both vertical and horizontal axis) [20–23], in compressor blades [24–26], rocket nozzles [27,28], and vehicles such as aircraft [5,29–31] and trucks [32].

These devices consist of two asymmetric electrodes separated by a dielectric film made of materials such as Kapton, PMMA or glass [33,34]. Voltage is applied to the top electrode, making it the active electrode. The bottom electrode is covered by the dielectric and is connected to the ground. Figure 1 shows a schematic of a DBD actuator.

A DBD works by applying a high voltage of the order of kV and a high frequency (hundreds to thousands of Hz) [34]. When the voltage is applied due to the intense electric field between the electrodes, the free electrons are accelerated. When these electrons collide with neutral molecules, they generate ionization on impact producing additional electrons and ions. The electrons released on impact accelerate and collide with more molecules, forming a chain reaction, which is known as the electron avalanche mechanism. Due to the collisions of the electrons and ions with neutral gas molecules, e.g., air, a net force transfers

an amount of momentum, which produces an induced flow that manifests itself as a wall jet. This net force is what gives DBD actuators their ability to control flow. When these electrons collide with neutral molecules, they generate ionization by impact, producing additional electrons and ions. The electrons released on impact accelerate and collide with more molecules, creating a chain reaction known as the electron avalanche mechanism. The collisions of the electrons and ions with neutral gas molecules, e.g., air, create a net force that transfers an amount of momentum that creates an induced flow that manifests itself as a wall jet. This net force is what gives DBD actuators their ability to control flow.

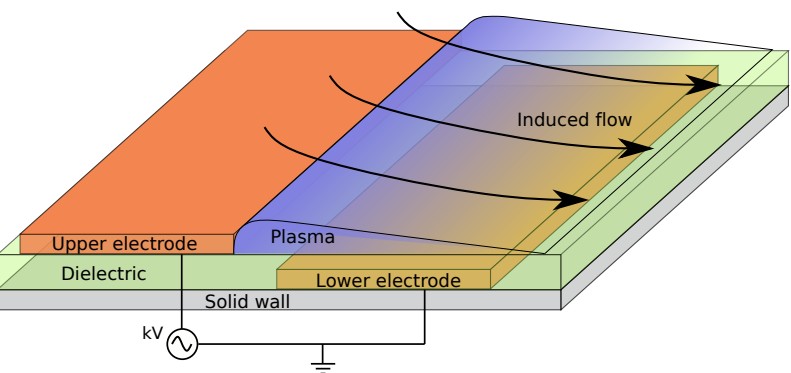

**Figure 1.** Schematic of a DBD actuator.

The study of the applications and the characterization of DBD actuators has been carried out by means of experimental techniques and with numerical simulations. The use of simulations to study DBD actuators plays a very important role in understanding and analyzing applications, which can be expensive or technically complex using experimental techniques. Simulations allow the detailed study of aspects such as the flow structure near the wall, turbulent structures, the interaction of the actuator when placed in moving elements such as turbomachinery, or under environmental conditions that are difficult to reproduce in the laboratory.

Several numerical models have been developed for the simulation of DBD actuators. The first-principles models are based on plasma physics and use the Boltzmann equation [35–38]. They are the most accurate, but their computational cost is extremely high because they require very small time steps and very fine numerical meshes, as a consequence of the spatio-temporal scales of the plasma, e.g., the time scales are on the order of $10^{-8}$ s to $10^{-9}$ s [39,40]. Due to their high computational cost, they are not practical for studying engineering applications.

The main requirement for the numerical study of flow control applications is to reproduce the macroscopic effects that a DBD actuator has on the flow, manifested as an ionic wind (wall jet). This can be achieved by the body force generated by the actuator. Phenomenological models have been developed for this purpose. The phenomenological models simplify much of the plasma physics, which significantly reduces their computational cost; their main purpose is to determine the electrohydrodynamic (EHD) force generated by a DBD actuator [39,41]. The EHD force represents the macroscopic body force resulting from the collisions of the charged particles with the neutral particles. When this force is introduced into the equations that model a flow, it is possible to reproduce the effects that the DBD exerts on the fluid.

The empirical models use experimental data of the velocity field to estimate the actuator force; since they are based on experimental data, they adequately reproduce the flow induced by the DBD actuator, but it is difficult to generalize them because they depend on the configuration of the DBD actuator used in the experiments.

The objective of this work is to compare the capability and performance of different DBD actuator models with experimental data. Two phenomenological models and one empirical model were evaluated. The phenomenological models are Shyy's [42] and Suzen's [43], which are the most widely used models of this type. The empirical model is the Dörr and Kloker model [44]. To evaluate these models, several simulations were

performed comparing the force, the maximum induced velocity and the velocity profiles at different frequencies and voltages. To contrast the numerical results with real data, the works of Forte et al. [45], Kotsonis et al. [46] and Tang et al. [47] were used as a reference cases. Therefore, the simulated DBD actuators have the dimensions (under the limitations of the models), voltage and frequency used in the reference cases. As the models showed discrepancies with the reference data, a second objective was set and minor and easy-to-implement modifications were proposed to improve the accuracy of the results generated by the models. The modified models are also compared with the results of the base models and the reference data.

The contents of this manuscript are presented as follows: the DBD actuator models and their implementation in OpenFOAM are described in Section 2; in Section 3 the software, numerical schemes and mesh are presented; and the results and discussion are presented in Section 4. The conclusions are given in Section 5.

## 2. DBD Actuator Models

### 2.1. Shyy Model

Among the phenomenological models, one of the simplest is the Shyy model [42]. In this model, the electric field is assumed to have its maximum value at the edge of the top electrode and its intensity decreases linearly with the distance between the top electrode and the dielectric surface, as shown in Figure 2, and the charge density is assumed to be constant. The main advantage of this model is its simplicity and low computational cost. The variation of the electric field $\boldsymbol{E}$ is given by the expression,

$$|\boldsymbol{E}| = E_0 - k_1 x - k_2 y, \tag{1}$$

where $E_0$ is the initial electric field, defined by Shyy as $E_0 = V_0/x_g$, with $V_0$ as the applied voltage and $d_e$ the distance between the electrodes. The problem with this expression is that is not suitable for DBD actuators with a zero gap. Therefore the following expression is used:

$$E_0 = \frac{V_0}{\sqrt{x_g^2 + t_u^2}}. \tag{2}$$

By considering $t_u$, the thickness of the upper electrode, it is possible to determine $E_0$ for DBDs with no gap between the electrodes, since $x_g^2 \approx x_g^2 + t_u^2$.

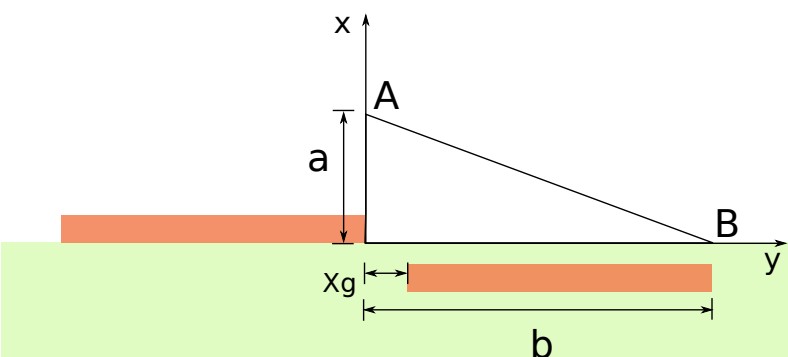

**Figure 2.** Linear distribution of the electric field in Shyy's model, where a denotes the height and b denotes the length of the electric field.

The constants $k_1$ and $k_2$ are determined by the condition that the intensity of the local electric field must be equal to the intensity of the breakdown electric field:

$$k_1 = \frac{E_0 - E_b}{b}, \tag{3}$$

$$k_2 = \frac{E_1 - E_b}{a}. \tag{4}$$

In the original formulation of Shyy, $a$ is the height of the plasma and is equal to 1.5 mm and $b$ is the width of the plasma and is equal to 1.5 mm. In this work, it is assumed that $a$ is equal to the height of the upper electrode $t_u$ and $b$ is equal to the length of the lower electrode $l_l$.

In Shyy's work, the breakdown electric field strength is $E_b = 3.0 \times 10^6$ V/m, this value corresponds to the dielectric strength of air. However, it is possible to obtain a better estimate of the value of the breakdown electric field for an atmospheric discharge by considering the empirical formula of Peek's law, which was originally developed for corona discharges, but its applicability to DBD actuators has been shown [40]:

$$E_b = E_a \delta \epsilon \left( 1 + \frac{0.308}{\sqrt{0.5 \delta t_u}} \right), \tag{5}$$

where $E_a = 3.1 \times 10^6$ V/m is the dielectric strength of air, $t_u$ is the thickness of the upper electrode, $\epsilon$ is a surface roughness factor is equal to 1 for smooth surfaces,

$$\delta = \frac{298p}{T},$$

is the relative atmospheric density factor, $T$ is the temperature in Kelvin, $p$ the pressure in atm. The value of $\delta \approx 1$ for air at standard conditions.

Then the EHD force is

$$f_b = \omega \Delta t n_c e_c E \delta_b, \tag{6}$$

where $\omega$ is the frequency of the applied voltage, $\Delta t = 67$ μs is the duration of the plasma discharge, $e_c = 1.602 \times 10^{-19}$ C is the charge of an electron, $n_c = 1.0 \times 10^{17}$ m$^{-3}$ is the number density, and $\delta_b$ is 1 if the electric field is above its critical value and 0 otherwise.

### 2.2. Suzen Model

The Suzen model [43] is derived from the Maxwell's and plasma physics concepts such as the Debye length and the Boltzmann's relation. Starting from Maxwell's equations,

$$\nabla \cdot \boldsymbol{D} = \rho_c, \tag{7}$$

$$\nabla \cdot \boldsymbol{B} = 0, \tag{8}$$

$$\nabla \times \boldsymbol{E} = -\frac{\partial B}{\partial t}, \tag{9}$$

$$\nabla \times \boldsymbol{H} = \boldsymbol{J} - \frac{\partial D}{\partial t}. \tag{10}$$

It is assumed that the charge redistribution is an instantaneous process, and the system is quasi-stationary [40,43]. Therefore the magnetic field $\boldsymbol{H}$, the current density $\boldsymbol{J}$, and the induction magnetic field $\boldsymbol{B}$ are close to zero, so they are depreciated. As a result of Maxwell's equations, only Gauss's law and Faraday's law remain. Faraday's law is rewritten as

$$\nabla \times \boldsymbol{E} = 0. \tag{11}$$

The Equation (11) implies that the electric field is irrotational, this allows the determination of the electric field as the gradient of a potential

$$\boldsymbol{E} = -\nabla \Phi, \tag{12}$$

where $\Phi$ is the electric potential.

The dielectric material is assumed to be linear, then the electric displacement field is linear:

$$\boldsymbol{D} = \varepsilon_0 \varepsilon_r \boldsymbol{E} \tag{13}$$

where $\varepsilon_0$ is the vacuum permittivity and $\varepsilon_r$ is the relative permittivity.

Replacing Equation (13) into Equation (7) gives

$$\nabla \cdot (\varepsilon_r \boldsymbol{E}) = \frac{\rho_c}{\varepsilon_0}. \tag{14}$$

There are two electric fields in the system—the external electric field generated by the potential difference at the electrodes, and the local electric field caused by the presence of charges above the surface. To deal with this, Suzen divided the electric potential $\Phi$ into an external and a local component:

$$\Phi = \phi + \varphi, \tag{15}$$

where $\phi$ is the electric potential due to the electrodes, and $\varphi$ is the potential generated by the charge density.

The external electric field is determined by means of a Laplace equation:

$$\nabla \cdot (\varepsilon_r \nabla \phi) = 0. \tag{16}$$

It is assumed that the plasma is in equilibrium, so that the Boltzmann relation can be used, which together with the definition of the Debye length gives

$$\varphi = -\frac{\rho_c \lambda_D^2}{\varepsilon_0}, \tag{17}$$

here $\lambda_D$ is the Debye length.

By replacing the electric field in Equation (14) with the gradient of the electric potential, and then replacing the potential with Equation (17) an expression for the charge density is obtained:

$$\nabla \cdot (\varepsilon_r \nabla \rho_c) = \frac{\rho_c}{\lambda_D^2}. \tag{18}$$

Solving Equations (16) and (18) allows the calculation of the EHD force with the following expression:

$$f_b = \rho_c \boldsymbol{E} = \rho_c (-\nabla \phi). \tag{19}$$

For Equation (16) at the electrodes, Dirichlet boundary conditions are applied, at the upper electrode $\phi = \phi(t)$ and at the lower electrode $\phi = 0$ at the far boundaries Neumann conditions $\partial \phi / \partial \boldsymbol{n} = 0$ are applied. For Equation (18) at the far boundaries and at the upper electrode $\partial \rho_c / \partial \boldsymbol{n} = 0$ is applied, the variation of the charge density on the surface of the dielectric is determined by the following function:

$$\rho_{c,w}(x, t) = \rho_c^{max} G(x) f(t), \tag{20}$$

where $\rho_c^{max}$ is the maximum charge density, and the density variation at the wall is,

$$G(x) = \exp\left[\frac{-(x - \mu_c)^2}{2\sigma^2}\right] \quad x \geq 0, \tag{21}$$

where $\mu$ is the location on the x-axis for the maximum value, and $\sigma$ is the scale parameter for the decay rate. In this work, these parameters have the following values: $\mu_c = 0.0$ m, so the maximum value will be at the edge of the top electrode, and $\sigma = 0.3$. The nondimensional boundary conditions for the model equations are shown in Figure 3.

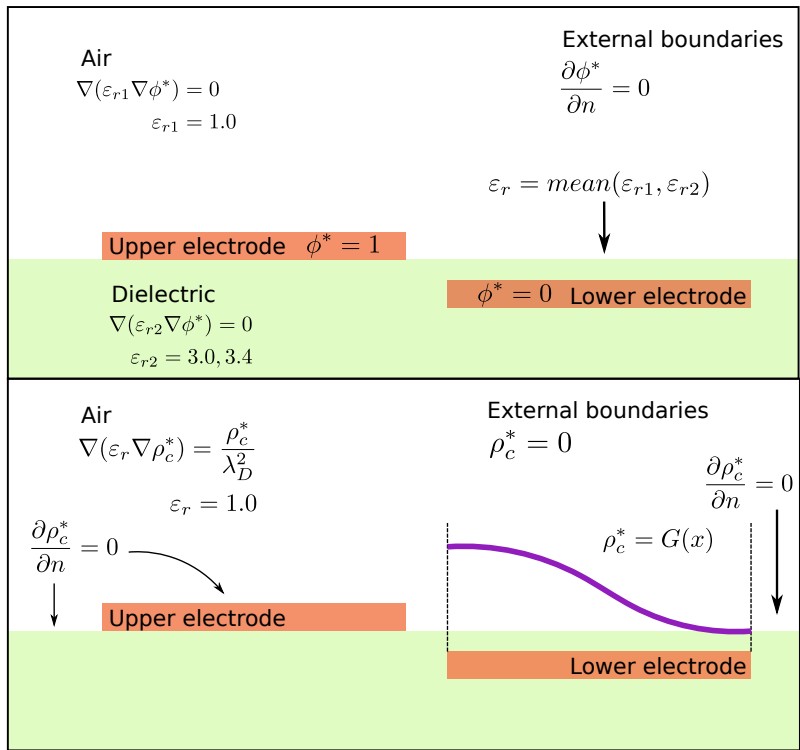

**Figure 3.** Boundary conditions for the Suzen model.

The time variation of the EHD force due to the voltage waveform signal is accounted for by $f(t)$:

$$f(t) = \sin(2\pi\omega t), \tag{22}$$

where $\omega$ is the AC frequency in Hz, ad $t$ is the time. In this work the same values used by Suzen are applied (base model): $\rho_c^{max} = 8.0 \times 10^{-4}$ C/m$^3$ and $\lambda_D = 1.0 \times 10^{-3}$ m.

The maximum charge density and the Debye length change with the applied voltage and frequency, in Suzen's model these values are calibrated for a specific voltage. If these variables are kept constant for all voltages and frequencies, the accuracy of the model is reduced. To account for the variation of these parameters, the relationship proposed by Wang et al. [48] is used, with Equation (23), the maximum charge density is obtained as a function of voltage; and with the relationship derived by Omidi and Mazaheri [49], Equation (24), the Debye length is obtained as a function of voltage and frequency. The modified Suzen model is obtained by applying both relations.

$$\rho_c^{max} = 4 \times 10^{-6} V^{2.241} \tag{23}$$

$$\lambda_D = 0.2 \left( \arctan \left( -170 f^{-5.124} \right) + 1.768 \right) \cdot \left( 0.3 \times 10^{-2} V - 7.42 \times 10^{-4} \right) \tag{24}$$

### 2.3. Dörr and Kloker Model

The Dörr and Kloker [44] model is empirical. This model is based on a velocity field approach. The velocity field generated by a DBD actuator is determined using anemometry techniques such as PIV or hot wire. Velocity field-based models use the Navier–Stokes equations to balance the momentum and determine the force, which is possible because the velocity field has already been solved experimentally. Typically, these models neglect pressure gradients and the y-axis force component, which has been shown to be much smaller than the x-axis force [50,51].

The Dörr and Kloker model consists of the force distribution parallel to the wall given by Maden et al. [52] and a factor to make this distribution dimensional. The following

equations describe the distribution and the dimensionless magnitude of the force along the x-axis and y-axis:

$$X(x) = \left(a_0 a_1 x + a_0^2 a_2 x^2\right)e^{-a_0 x}, \tag{25}$$

$$Y(y) = \left(b_1 y + b_2 y^2\right)e^{-b_0 y^{2/5}}. \tag{26}$$

To obtain the distribution of the force, the scalar multiplication of these polynomials must be carried out in such a way that

$$f(x,y) = c_x\left(a_0 a_1 x + a_0^2 a_2 x^2\right)e^{-a_0 x}\left(b_1 y + b_2 y^2\right)e^{-b_0 y^{2/5}}, \tag{27}$$

where $x, y \geq 0$; $a_0, b_0 > 0$; $a_{1,2}, b_{1,2}, c_x \ni \mathbb{R}$. These coefficients make it possible to control the distribution of the force, allowing the modification of the length and height of the force. Table 1 lists the coefficients used by Dörr and Kloker, the force distribution with these coefficients is shown in Figure 4.

**Table 1.** Coefficients for the Dörr and Kloker model.

| Coefficient | $a_0$ | $a_1$ | $a_2$ | $b_0$ | $b_1$ | $b_2$ | $c_x$ |
|---|---|---|---|---|---|---|---|
| Value | 55 | 8 | 10 | 34 | 2.7 | 0.7 | 80 |

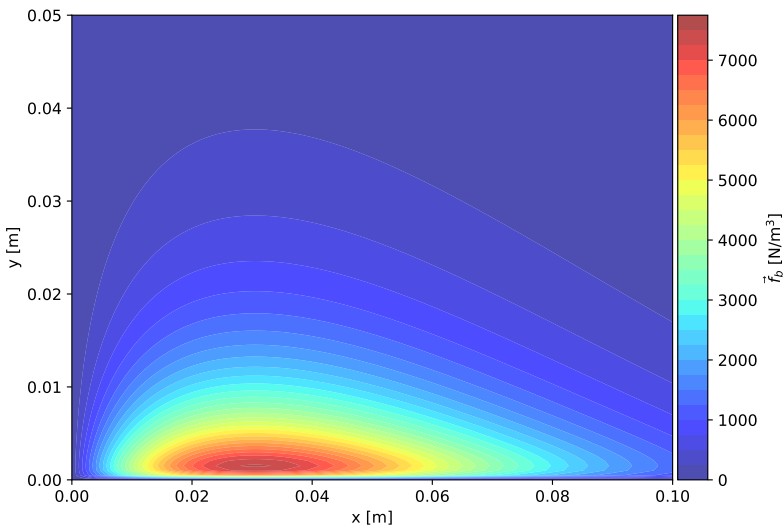

**Figure 4.** Nondimensional force distribution for the Dörr and Kloker model.

As mentioned, the Equation (27) produces a dimensionless force distribution. Therefore, to obtain a dimensional force, Dörr and Kloker multiplies it by a dimensional factor:

$$f_b(x,y) = f(x,y)\frac{\bar{\rho}_\infty \bar{U}_\infty^2}{L} = f(x,y)\frac{\bar{\rho}_\infty^2 \bar{U}_\infty^3}{Re\bar{\mu}_\infty}, \tag{28}$$

where $\rho_\infty$ is the air density, $U_\infty$ is the free stream velocity, $L$ is the length reference and $Re$ is the Reynolds number.

From the dimensional factor of Equation (28) it is evident that there is a problem with its use, because no link to the operating parameters of a DBD actuator (voltage and frequency), and also takes into account the free stream velocity, then this is not valid in cases with quiescent air, this limits the use of this model. Therefore, in this work, a technique that our research group previously established (Bernal-Orozco et al. [53]) was applied. This technique consists of multiplying the dimensionless distribution from Equation (27) by

the value of the maximum EHD force from the data of Hofkens [54], which allows the obtaining of a better approximation of the force for a given voltage and frequency.

In addition to the use of the above-mentioned technique, the coefficients of the model were modified unlike previous work in which the original coefficients were used, because if the coefficients given by Dörr and Kloker are used, the force will have a length of 10 cm and a height of 3 cm (see Figure 4) which is significantly larger than what would be obtained with a DBD actuator with the configurations of the cases used as a benchmark for this work. Therefore, in the present work, it is proposed that the length of the force should be equal to the length of the lower electrode and that its height is between 1 mm and 2 mm. Table 2 shows the modified coefficients for cases 1 and 2.

**Table 2.** Coefficients for the modified Dörr and Kloker model.

| Coefficient | Case 1 | Case 2 |
|:---:|:---:|:---:|
| $a_0$ | 450 | 180 |
| $a_1$ | 8 | 8 |
| $a_2$ | 10 | 10 |
| $b_0$ | 85 | 85 |
| $b_1$ | 2.7 | 2.7 |
| $b_2$ | 0.7 | 0.7 |
| $c_x$ | 2500 | 2500 |

## 3. Numerical Setup

The airflow is simulated using the incompressible Navier–Stokes equations. Within the momentum equation, the EHD force is introduced as a source term to produce the effects of the DBD actuator over the flow:

$$\nabla \cdot \boldsymbol{u} = 0, \tag{29}$$

$$\frac{\partial \boldsymbol{u}}{\partial t} + (\boldsymbol{u} \cdot \nabla)\boldsymbol{u} = -\frac{1}{\rho}\nabla p + \nu\nabla^2\boldsymbol{u} + \frac{f_b}{\rho}, \tag{30}$$

where $f_b$ is the EHD force in N/m$^3$, the force is previously determined by the DBD models.

The simulations were performed in OpenFOAM 8.0. A custom-made solver was used for the Suzen model, the Shyy and Dörr and Kloker models were implemented using the fvOptions tool. The flow solver for all cases was pimpleFoam, configured in its PISO mode. The turbulence model was the k-$\omega$SST model. Second-order schemes were used. A preconditioned conjugate gradient solver was used for the symmetric matrices and a smooth solver for the asymmetric matrices. The mesh is structured, and for the Suzen model cases consists of two subdomains as shown in Figure 5.

A mesh convergence study was performed using the Richardson method and following the guidelines of Celik et al. [55], three meshes were tested: coarse, medium and fine. The mesh convergence study was performed with an actuator with dimensions corresponding to case 1 at 16 kVpp and 2 kHz, it was determined that the fine mesh is adequate. The fine mesh was used as a reference to generate the meshes for the other cases. Table 3 shows the variables for the mesh convergence study, in Table 4 the mesh convergence results are presented, two quantities of interest were observed, the maximum induced velocity $U_{max}$ and the integral force $f_b^{Tot}$, the grid convergence index $GCI_{21}$ was 0.1614% and 0.0102%, respectively, therefore the error with the fine mesh is sufficiently low.

**Table 3.** Variables for the mesh sensitivity study.

| Mesh | Cells | h [m] | $y_{max}^+$ | $U_{max}$ [m/s] | $f_b^{Tot}$ [N/m] |
|:---:|:---:|:---:|:---:|:---:|:---:|
| 1 Fine | 9934 | $6.67 \times 10^{-6}$ | 0.00453083 | 1.5789 | 0.004449 |
| 2 Medium | 5616 | $1.0 \times 10^{-5}$ | 0.00331336 | 1.5662 | 0.004441 |
| 3 Coarse | 2164 | $1.5 \times 10^{-5}$ | 0.0171494 | 1.4738 | 0.004264 |

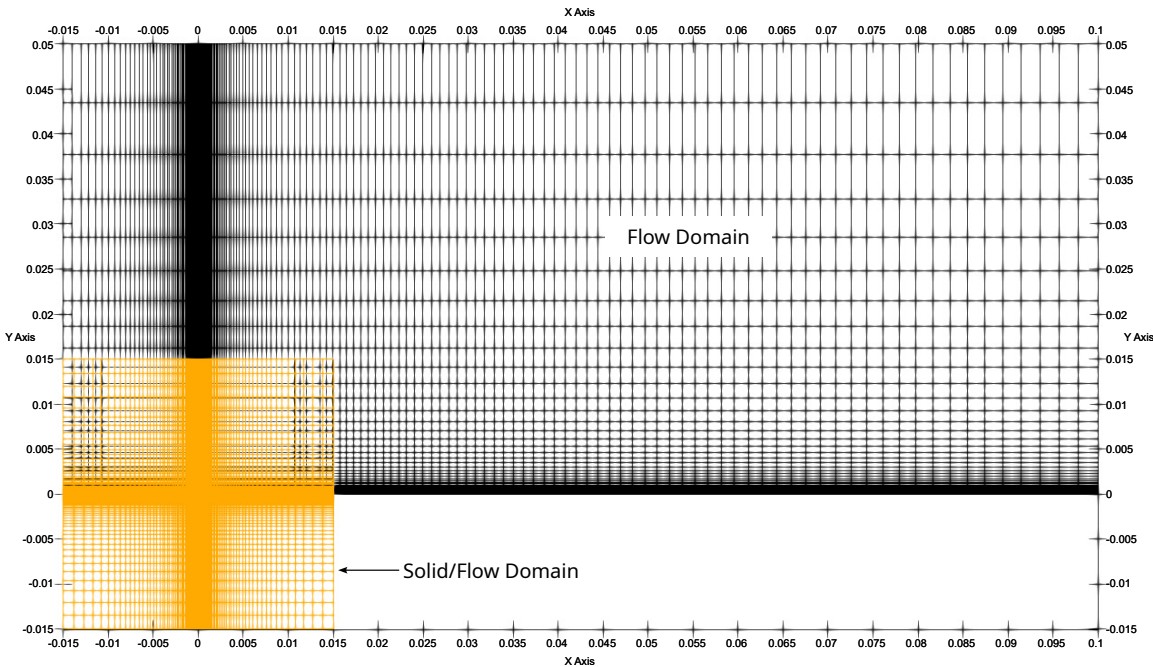

**Figure 5.** Mesh for the electric field (orange) and fluid mesh (black).

**Table 4.** Mesh sensitivity study.

| $\phi$ | $r_{21}$ | $r_{32}$ | Convergence | $p$ | $\phi_0$ | $e_{21}$ [%] | $e_{21}^{ext}$ [%] | $GCI_{21}$ [%] |
|---|---|---|---|---|---|---|---|---|
| $U_{max}$ | 1.4993 | 1.5 | Monotonic | 4.88 | 1.5810 | 0.8058 | 0.1290 | 0.1614 |
| $f_b^{Tot}$ | 1.4993 | 1.5 | Monotonic | 7.69 | 0.0044 | 0.1758 | 0.0082 | 0.0102 |

## 4. Results and Discussion

The data from Kotsonis, Tang, and Forte was used as a benchmark, this implies that the simulated cases have the dimensions, voltage, and frequency parameters used in those works. Table 5 shows the configuration used for each of the cases. Results are presented for the standard version of the Shyy and Suzen models and for their modified versions. In the case of the Dörr and Kloker model, only the results of the modified version are presented, because the dimensional factor is not a function of voltage and frequency, which makes comparison impossible. Please note that all the voltages in this section are peak to peak voltages. The air is quiescent in all the simulations, the flow effects are the results of the models action. The error bars of the experimental data are plotted, where available.

**Table 5.** Parameters for the simulation cases.

| Parameters | Case 1 | Case 2 | Case 3 |
|---|---|---|---|
| Voltage [1] | 8 kV to 16 kV | 14 kV to 19.5 kV | 16 kV to 52 kV |
| Frequency | 2 kHz | 2 kHz | 1 kHz |
| $L_u$ | 10 mm | 15 mm | 10 mm |
| $L_l$ | 10 mm | 25 mm | 20 mm |
| $t_u$ | 60 μm | 50 μm | 100 μm |
| $t_l$ | 60 μm | 50 μm | 100 μm |
| $t_d$ | 110 μm | 330 μm | 2 mm |
| $X_g$ | 0 | 0 | 0 |
| $\varepsilon_r$ | 3.4 | 3.4 | 3.0 |
| Reference | Kotsonis et al. [46] | Tang et al. [47] | Forte et al. [45] |

[1] The voltages are given as peak-peak voltage.

Case 1 allows the comparison of the velocity field structure with the force distribution, as well as the relationship between force components and voltage. Case 2 studies the force as a function of voltage at two different frequencies, and velocity.

### 4.1. Case 1 Results

Figures 6–8 show the velocity and force fields for each model, where the reference wall jet and the force field are projected over the numerical fields. The black segmented lines are the force distribution; the purple segmented line is the shape of the wall jet and the dotted lines with arrows are the streamlines, the jet has a maximum velocity of 3.0 m/s a length of 10 mm and a height of nearly 1 mm.

Figure 6 shows the velocity and force field for the modified Shyy model. With the modification applied to the model, the wall jet has a height of about 0.5 mm which is half of the reference, and an approximate length of 7 mm, making it the smallest jet obtained with the models analyzed in this work. However, the velocity is close to 3.0 m/s, which is the velocity reported in the reference case (at 12 kV and 2 kHz). The force has the characteristic triangular shape of this model, and its maximum intensity is located near the edge of the upper electrode. It is possible to calibrate the model to extend the jet length by increasing either *b* or *a*, which would increase the force, although experimental data on plasma extension is needed to do this properly. Without modifications, the velocity can reach 48 m/s which is excessive for a DBD actuator, this high velocity is the result of the larger volume force since it has a height of 1.5 mm then the total force is greater than that from the modified version of the model.

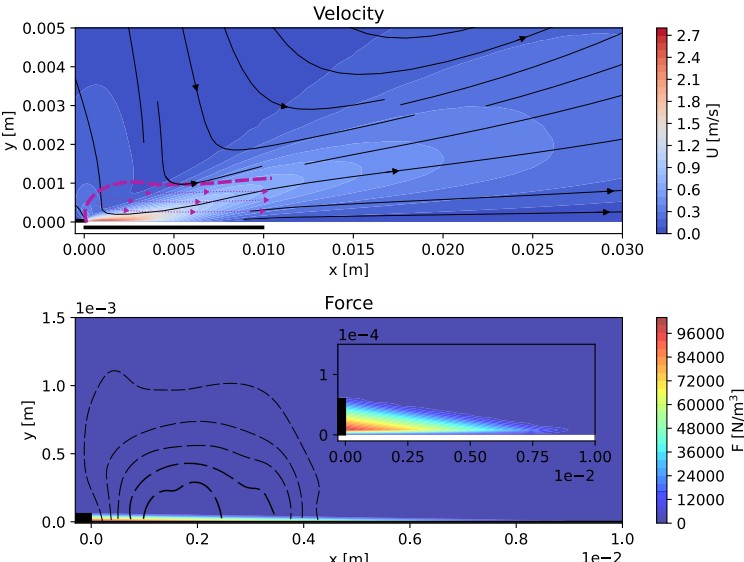

**Figure 6.** Modified Shyy model velocity field (**top**) and force (**bottom**) at 12 kV and 2 kHz. Purple lines and arrows are the reference wall jet, black arrows are the streamlines of the numerical results.

The results obtained with the Suzen model are shown in Figure 7. In the upper part, the flow field is observed, the wall jet has an approximate length of 25 mm and a height of about 1.5 mm, the maximum velocity is 1.4 m/s, which is lower than the 3.0 m/s from the benchmark case for this frequency and voltage. For the force, it is observed that its maximum magnitude is at the lower right corner of the upper electrode, this point corresponds to the place where the electric field has its highest intensity, and the charge density on the dielectric surface has its peak near this point. With this model, the force distribution has the minimum extension among the tested models, then the total force is lower and consequently the achieved velocity is lower. For the standard Suzen model, the structure of the velocity field is similar, but the maximum induced velocity is 1.32 m/s, which means that this velocity is almost half the velocity of the reference case and lower than that of the improved model.

The velocity field and the force for the Dörr and Kloker model are shown in Figure 8. The jet has a length of 33 mm and a height of 2.5 mm, which is similar to the Suzen model and the velocity field measured by Kotsonis et al. [46]. The maximum velocity is 3.15 m/s, which is closer to the reference velocity than the results obtained with the Shyy model.

By modifying the model coefficients, the force has a length of 10.0 mm, which is equal to the length of the lower electrode, and a height of 2.0 mm, the maximum value of the force is located between x = 30.0 mm and x = 45.0 mm, the location of the maximum value of the force influences the velocity profiles. Although the force distribution has a larger extension than that obtained with the Shyy and Suzen models, the wall jet dimensions and the maximum velocity adequately match the reference.

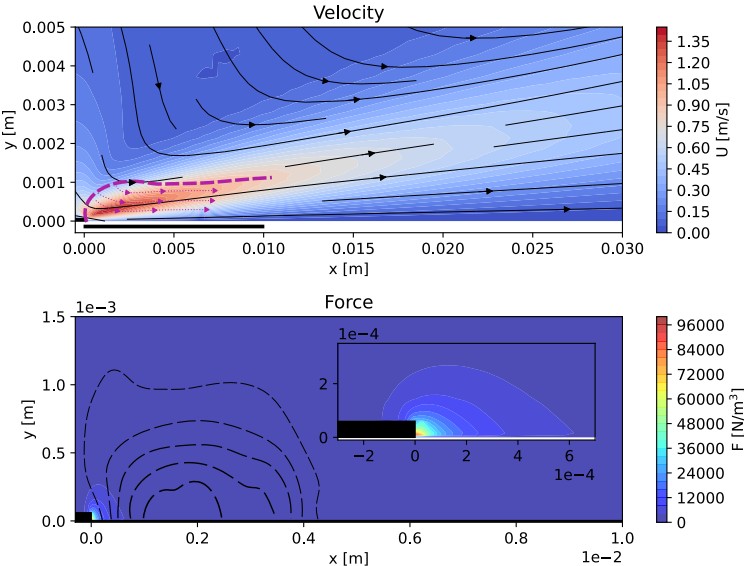

**Figure 7.** Unmodified Suzen model velocity field (**top**) and force (**bottom**) at 12 kV and 2 kHz. Purple lines and arrows are the reference wall jet, black arrows are the streamlines of the numerical results.

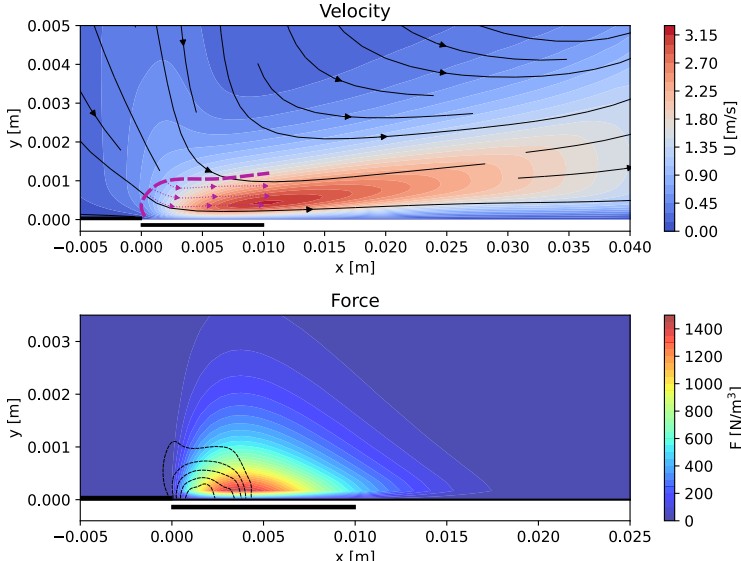

**Figure 8.** Modified Dörr and Kloker model velocity field (**top**) and force (**bottom**) at 12 kV and 2 kHz. Purple lines and arrows are the reference wall jet, black arrows are the streamlines of the numerical results.

Under the Case 1 configuration, the force components are compared for a range of voltages from 8 kv to 16 kv. The force components for the unmodified Shyy and Suzen models are shown on the left side of Figure 9 and those obtained with the modified models on the right side. The force of Kotsonis follows a potential relation $f_b \propto V^n$ that has been reported in several studies [46,56,57]. In general, the magnitude of $f_y$ has less influence on

the induced flow compared to $f_x$. The standard Shyy model overestimates the force over the entire voltage range and has a linear behavior. For the standard Suzen model, the trend is also linear, and below 10 kV the force is overestimated, above this value the magnitude of the force is below that of the reference. With the modified Shyy model, the force magnitude is reduced for both force components, but the x-component is still overestimated and the linear tendency is maintained. With the improved Suzen model, the x-component shows a good approximation at 8 kV and 10 kV, at the positions marked with black arrows; for higher voltages, the force is below the reference values, but the behavior follows a potential law. The force from the Dörr and Kloker model can be fitted to a logarithmic trend, the force is quite close to the reference value at 8 kV, for the other voltages it overestimates the force and agrees with Shyy's model at 10 kV and 12 kV.

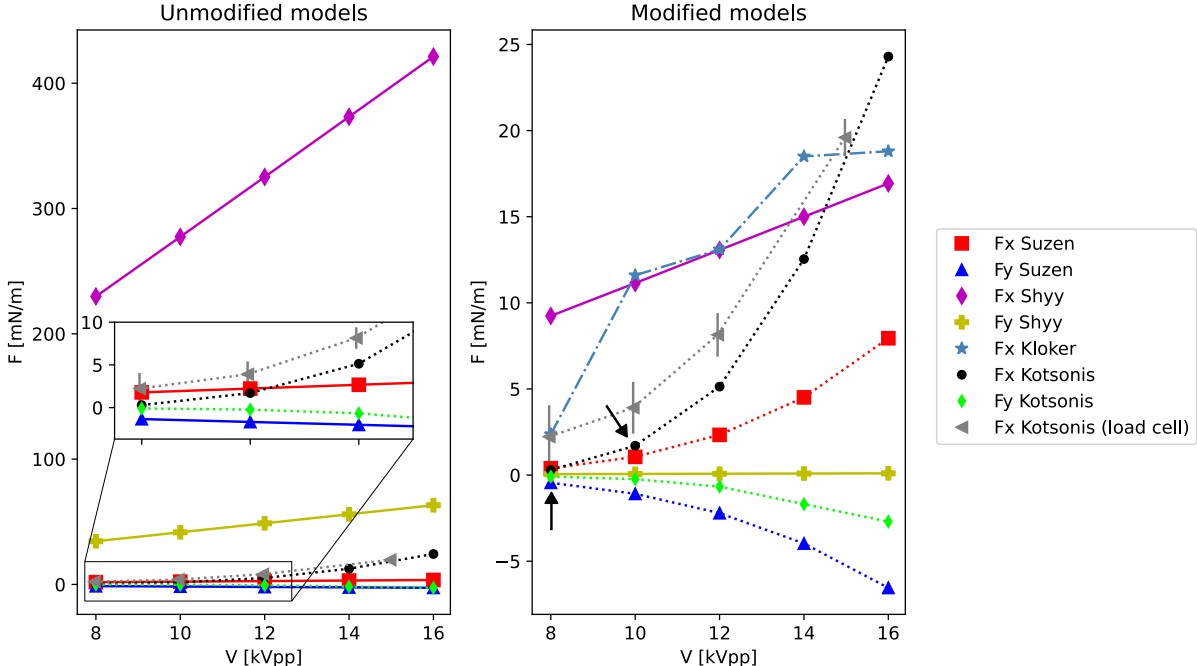

**Figure 9.** Effect of voltage at 2 kHz (case 1 setup) on the force components, unmodified models on the left, modified models on the right. The black arrows show where the numerical results are close to the reference value.

*4.2. Case 2 Results*

In Figure 10, the effects of voltage and frequency on the force are compared (case 2 configuration), on the left are the results for the unmodified models and on the right are the results obtained with the modified models. The black arrows indicate where the numerical results are close to the validation data. Tang's results (reference data) show that the force increases with voltage following a logarithmic trend, and that increasing frequency produces an increase in force.

For the unmodified Suzen model, the increase in frequency does not change the magnitude of the force, while for the modified Shyy model, there is an increase in force when a higher frequency is applied. At 14 kV and 1 kHz the Suzen model is close to Tang's results. With the modified Suzen model when using Equation (24) the frequency is taken into account in the calculation of the Debye length, then there is a difference between the force for 1 kHz and 2 kHz; however, the difference is minimal compared to the reference, it is noteworthy that the force for the case at 2 kHz is close to the reference at 14 kV, 18 kV and 20 kV. The force calculated with the modified Shyy model is lower than that of the unmodified Shyy model, but is still larger than the reference. For the Dörr and Kloker model, the force is greater than that of the reference, moreover, the dimensional factor depends on the data reported by Hofkens,

which is available for a range of 8 kV to 16 kV at 2 kHz, this limits the evaluation of this model at 1 kHz, by extrapolation a value for 18 kV has been estimated.

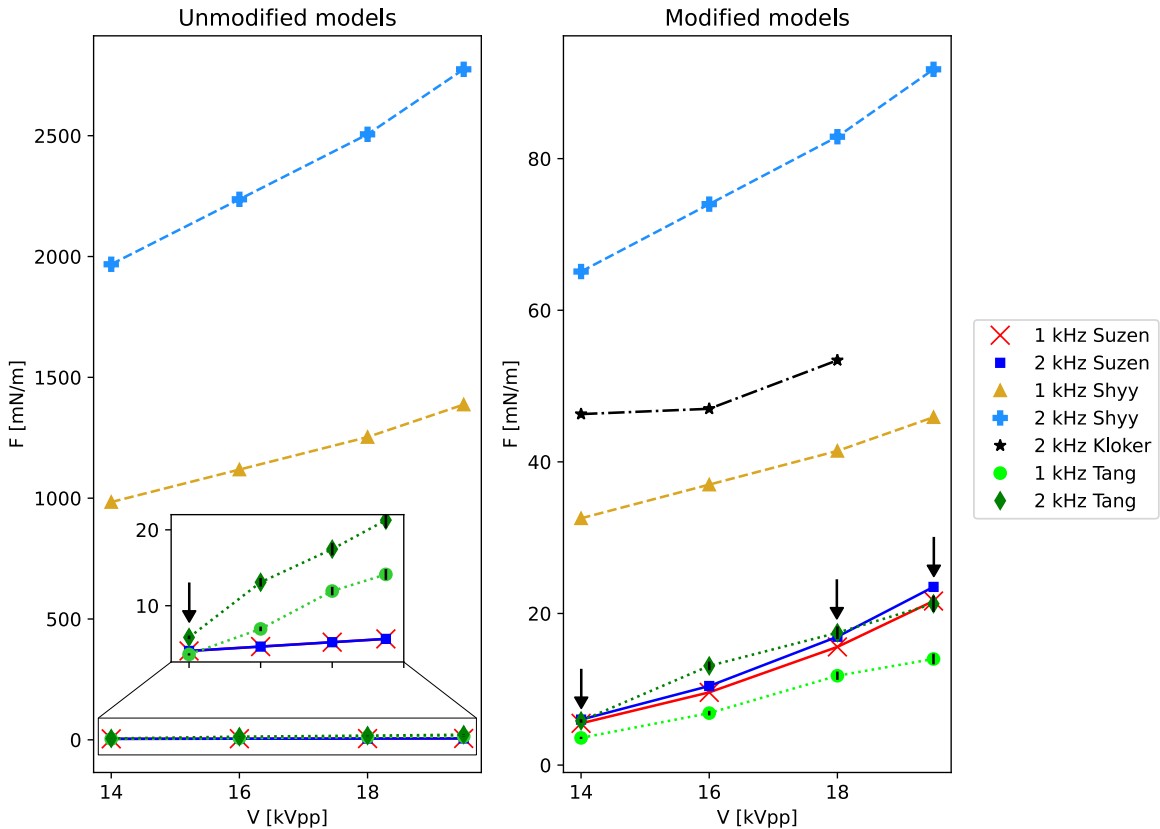

**Figure 10.** Effect of voltage and effect of frequency (case 2 setup) on the force, unmodified models on the left, modified models on the right. The black arrows show where the numerical results are close to the reference value.

The velocity profiles for the unmodified Shyy and Suzen models, measured at x = 10 mm, 15 mm, 40 mm and 75 mm along the x-axis, are shown in Figure 11, the actuators have the case 2 configuration at 18 kV and 2 kHz. As a consequence of the underestimation of the force by the Suzen model, the velocity profiles show a lower velocity than the real case. For the velocity profile measured at x= 10 mm, there is a velocity difference of almost 3.0 m/s. With Shyy's model, since the integral force is larger than it should be, the velocity is extremely high for a DBD actuator. The velocity reaches more than 60 m/s, which is wrong.

The velocity profiles for the modified models are shown in Figure 12. The profiles for the Suzen model are better approximated than those from the standard version, the wall jet thickness is well reproduced, and the difference between the numerical maximum velocity and the reference velocity is smaller, such that for the x = 10 mm profile there is a difference of 1.3 m/s, which is a smaller difference than the one from the standard Suzen model. With the modified Shyy model, the velocity is significantly lower than with the unmodified model, which is a substantial improvement, for the profile at x = 10 mm there is a difference of about 1.2 m/s, the profiles at x = 40 mm and 75 mm show that the wall jet dissipates, indicating that the wall jet length is shorter. For the Dörr and Kloker model, all velocity profiles have a higher velocity than that reported by Tang (for each corresponding position), it can be seen that the velocity profiles at x = 15 mm and 40 mm have a velocity higher or close to that of the profile at x = 10 mm, while one would expect the velocity to decrease with increasing distance from the DBD actuator, as shown by Tang's data. This discrepancy in the order of the velocity profiles is explained by looking at Figure 8, where

the force distribution has the zone of highest intensity approximately between x = 30 mm to 45 mm, unlike the other models, and what happens with a real DBD actuator where the point of highest force intensity is near the edge of the top electrode near x = 0. Therefore, as the point of maximum force is shifted, the formation of the wall jet is delayed and the velocity profiles behind the point of maximum force have a higher velocity than the profiles in front of it.

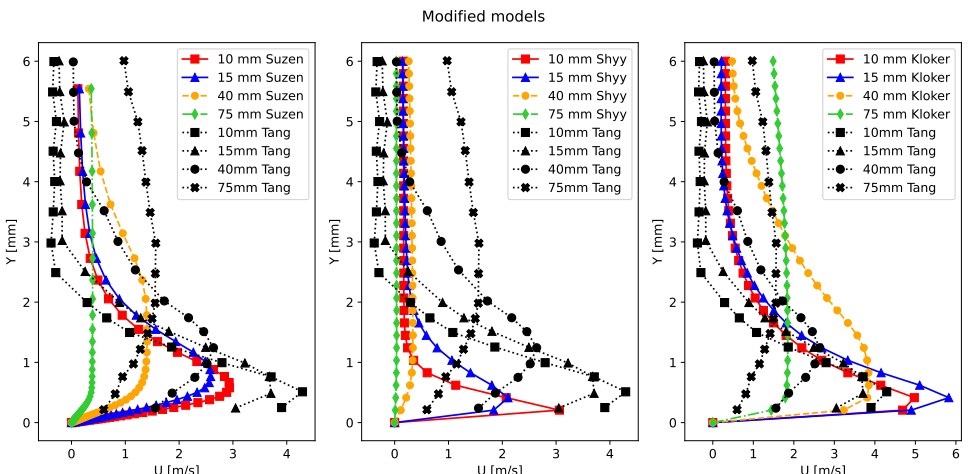

**Figure 11.** Velocity profiles for the unmodified models (case 2 setup) at 18 kV and 2 kHz.

**Figure 12.** Velocity profiles for the modified models (case 2 setup) at 18 kV and 2 kHz.

### 4.3. Case 3 Results

The maximum velocity as a function of voltage was analyzed using the case 3 configuration. Figure 13 on the left side shows the results for the unmodified models, on the right side the results obtained with the modified models, the reference data are taken from Forte et al. [45] and the reference data can be fitted to a logarithmic trend. For the unmodified Shyy model, the velocity is significantly higher than that reported by Forte, and the trend is linear. With the modified version of the Shyy model, when the plasma height is equal to

the thickness of the top electrode, the integral force is lower, resulting in an induced flow with a lower velocity than with the original model, but the magnitude of the velocity is still higher than the reference and the linear tendency is maintained. With the unmodified Suzen model, at 24 kV the maximum induced velocity is closer to the reference value. Below this voltage, it is higher than the experimental velocity, and above it, the velocity is lower than the experimental velocity, which is a consequence of the underestimation of the force by this model. Using values of maximum charge density and Debye length as a function of voltage and frequency, the Suzen model is improved and the velocity obtained with the model approaches the reference values at 16 kV, 20 kV, 24 kV, 28 kV, 32 kV and 36 kV, but at voltages above 36 kV, the magnitude of the experimental velocity is exceeded, and at even higher voltages, the deviation increases.

For case 3, the comparison of the Dörr and Kloker model is not included because although the coefficients can be adjusted so that the extension of the force has dimensions that fit the configuration of case 3. The same cannot be done with the magnitude of the force, since it depends on the dimensional factor. The dimensional factor data are available only for a frequency of 2 kHz, the experimental data for case 3 are at 1 kHz, and the only models that can be set to any frequency are the Shyy and Suzen models. If the Dörr and Kloker fitted with the available data for 2 kHz is included in this case, its results would be misleading. For an adequate comparison, the maximum force data at a frequency of 1 kHz is required to use it as a dimensional factor.

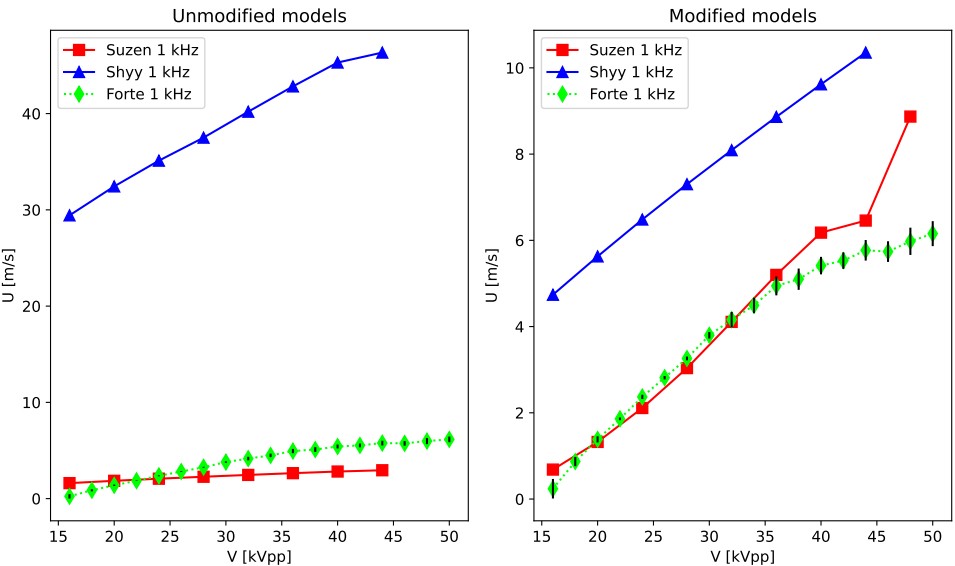

**Figure 13.** Maximum induced velocity as a function of the voltage at 1 kHz (case 3 setup).

In the velocity profiles for the unmodified models in the configuration of case 3, Figure 14, a behavior similar to that observed in case 2 is presented; the Suzen model underestimates the maximum velocity by more than 3.0 m/s and, on the contrary, and the Shyy model excessively overestimates the maximum reference velocity. An improvement is achieved with the modified models, as observed in Figure 15. The modified Suzen model reproduces the height of the wall jet and in this case overestimates the maximum velocity by 1.0 m/s; for the modified Shyy model, the average velocity profile at x = 10 mm is underestimated by 1.0 m/s and the wall jet is lower in height.

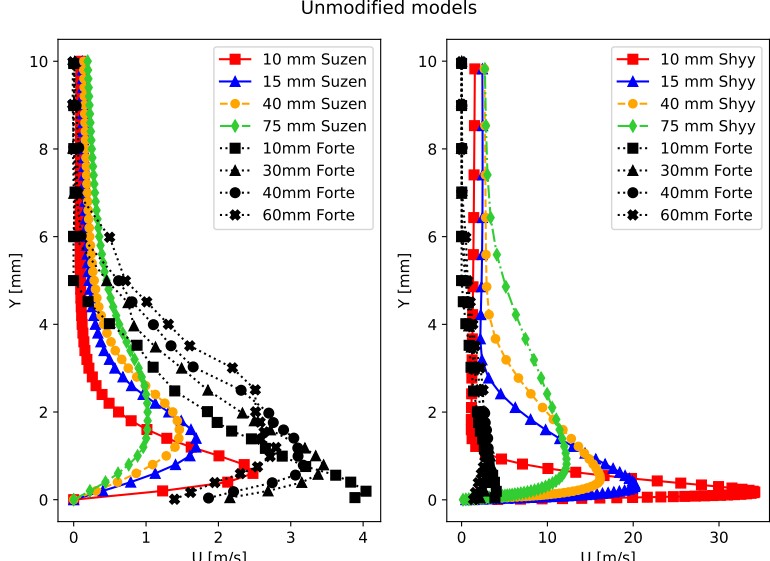

**Figure 14.** Velocity profiles for the unmodified models (case 3 setup) at 20 kV and 1 kHz.

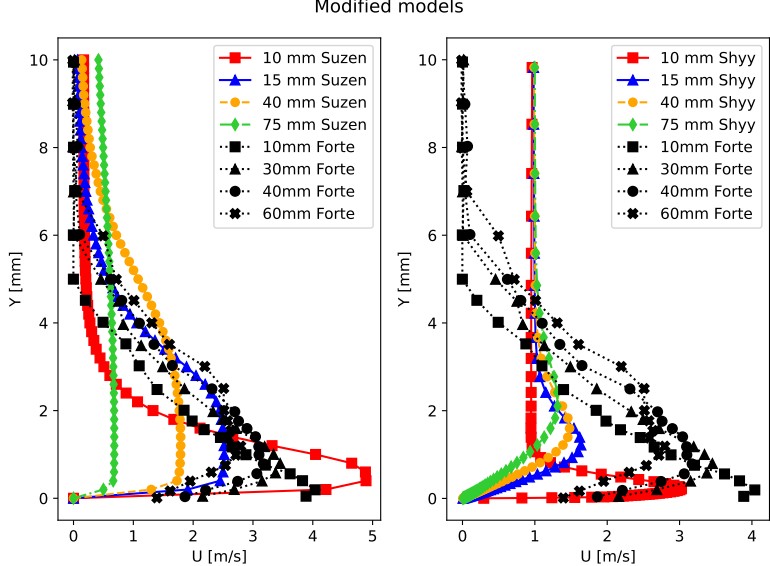

**Figure 15.** Velocity profiles for the modified models (case 3 setup) at 20 kV and 1 kHz.

### 4.4. Discussion

Despite modifications to the models, some discrepancies persist, with the most notable ones related to the force, as seen in Figures 9 and 10. Additionally, the Suzen model underestimates the velocity, while the Dörr and Kloker model overestimates it. It is possible to achieve a better fit by calibrating the models, as suggested below:

- To calibrate the Suzen model, the maximum charge density can be adjusted to bring the velocity profiles closer to the reference value. This is not an optimal procedure, but allows for fine tuning.
- For cases where data are available to use the Dörr and Kloker model, small corrections can be made by shifting the initial position of the force upstream over the x-axis so that the jet forms in the correct location. Since the velocity is overestimated, applying a lower force will make it lower, this can be achieved by reducing the length of the force to 0.75–0.8 of the lower electrode ($L_l$) length.

One aspect that should be mentioned is that a quantitative comparison of the force from the models with that of the reference, is affected by the inherent uncertainty from the experiments, as well as that attributed to the assumptions made when the force is determined using a technique based on the velocity field and the balance of the momentum equation, e.g., the assumption that the pressure gradient in the momentum equation is negligible [34,50]. This limits the accuracy of a quantitative comparison between the force from the models and the reference. Nevertheless, it is still possible to make a quantitative comparison, but it should be noted that the order of magnitude between the reference force and the one obtained from the models is consistent and that the variation does not exceed an arbitrary tolerance limit that can be set according to the required accuracy.

Despite the limitations to quantitative comparison of the force, the results shown in Figures 9 and 10 provide insight into the effect of the applied voltage on the force. The force from the models follows a linear trend, while the reference shows nonlinear behavior. In the case of Figure 9, the $f_x$ component reported by Kotsonis fits the relationship $f_x \propto V^\alpha$. From Figure 10, it can be seen that the base Suzen model does not consider the effect of frequency on the force, and the modified version of the model shows a slight change between the force at 1 kHz and 2 kHz, but the discrepancy is still significant. Therefore, these results show the discrepancy between the model and the experimental reference data.

To determine if the studied models are applicable for studying flow control cases, one must consider that the primary factor for controlling flow is the maximum velocity. Second, for flow separation cases, it is important to take into account the location within the boundary layer where the maximum velocity occurs. Chen [58] asserts that $U_{max}$ is the most important factor. In his work, a first-principles model was used to simulate a DBD actuator and it was compared against a phenomenological model. Despite differences in EHD force, both models were successful in controlling the flow separation. A similar observation about the importance of the location, dimensions, and maximum velocity of the wall jet on the ability to control the flow is made in Bernal-Orozco et al. [53].

In this sense, the results obtained in this study indicate that, despite the discrepancies in the EHD net force, it is demonstrated that a better approximation is achieved when observing the induced flow. From Figures 7 and 8, is can be observed that the wall jet has a similar thickness and length compared with the reference, and the velocity profiles (see Figures 12 and 15) show a similar result in the thickness of the wall jet for the modified Suzen and Dörr and Kloker models. Even though there is a difference between maximum velocity of $\pm 1.0$ m/s, this can be reduced by adjusting the maximum charge density in the Suzen model or reducing the extension and intensity of the force in the Dörr and Kloker model as indicated in previous guidelines. Returning to the idea that flow control depends primarily on maximum velocity, the results from Figure 13 show that the Suzen model should be able to control flow.

In summary, since the net force is not the main factor to allow the flow control, even though the wall jet is created by this force, the modified models achieved a better approximation regarding the induced flow and the velocity. Therefore, the models are suitable for use to study flow control applications. Moreover, they can be tuned to achieve a maximum velocity closer to the reference value.

## 5. Conclusions

This work has two objectives. First, to evaluate three numerical models for DBD actuators among themselves and with experimental data. The models were tested in three different cases, for different voltages and frequencies. The models studied were the Shyy model, the Suzen model, both of a phenomenological type, and the empirical Dörr and Kloker model. Large discrepancies were found between the models and the experimental data, leading to the second objective of modifying the models to reduce the discrepancies. The modified models were also compared with the base models and experimental data.

In the modified Shyy model, the height of the plasma is assumed to be equal to the thickness of the exposed electrode, which reduces the total force and leads to lower velocities.

Discrepancies, although much smaller than in the base model, are observed in the velocity profiles of cases 2 and 3. The velocity is lower than the reference value and the wall jet has a smaller height. The force against voltage remains higher and the linear trend is maintained. The modified Suzen model achieves better results. The height and length of the jet are very close to the reference values, but the velocity is underestimated by approximately 1.0 m/s in the velocity profiles of case 2, and overestimated by approximately 1.0 m/s in the profiles of case 3. For the case 3 configuration in the range of 16 kV to 36 kV, the maximum velocity $U_{max}$ is very close to the reference value. The force fits well for the case 2.

With the modified Dörr and Kloker model, it is possible to relate the force with a given voltage and frequency. A wall jet with a height close to the reference value is achieved, and in the velocity profiles, the height is close but still higher than the reference value. Additionally, the wall jet is formed downstream of the x-axis. Overall, the model achieves good results and it has a low computational cost, but it depends on the availability of experimental data for the EHD force dimensional factor, which limits its use.

Of the three models evaluated, the most robust one is the modified Suzen model because it can be used without the need for external data. By solving Gauss's law for the electric field, it takes into account part of the real physics of the problem. In addition, the induced flow presents a good approximation to the reference. Additionally, this is supported by the results of Figure 13, since the maximum velocity $U_{max}$ is the main factor that allows the model to control the flow, for some cases a calibration step may be required, to obtain $U_{max}$ closer to the reference. This can be achieved by tuning following the advises given in Section 4.

To further improve the models in a precise and systematic way, experimental data or first-principles numerical models are required, as planned for future work. With these, a database of coefficients and scaling factors could be generated for the Dörr and Kloker model. This would allow for better relationships for the maximum charge density and Debye length, and to set better boundary conditions for charge density in the Suzen model.

**Author Contributions:** Conceptualization, R.A.B.-O. and O.M.H.-C.; methodology, R.A.B.-O.; validation, R.A.B.-O.; analysis, R.A.B.-O.; investigation, R.A.B.-O. and I.C.-M.; writing—original draft preparation, R.A.B.-O. and I.C.-M.; writing—review and editing, R.A.B.-O., I.C.-M. and O.M.H.-C.; supervision, I.C.-M. and O.M.H.-C.; project administration, I.C.-M. and O.M.H.-C.; funding acquisition, I.C.-M. and O.M.H.-C. All authors have read and agreed to the published version of the manuscript.

**Funding:** This research was funded by the Technological Development Projects or Innovation at Instituto Politecnico Nacional (IPN), grant number SIP-20227009.

**Data Availability Statement:** The data that support the findings of this study can be available from the corresponding author upon reasonable request.

**Conflicts of Interest:** The authors declare no conflict of interest.

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
