# Peer review of "Performance of DBD Actuator Models under Various Operating Parameters and Modifications to Improve Them"

_fluids, doi:10.3390/fluids8040112_

Round 1

Reviewer 1 Report

In this paper, the performance of different numerical model, including the phenomenological Shyy and Suzen models and the empirical Dörr & Kloker model to simulate the velocity field and body force induced by dielectric barrier discharge (DBD) plasma actuator is compared. And all these models are modified for improvement. Results show that, these modifications are effective. This work is meaningful to the numerical investigation of the DBD-based plasma flow control. The current reviewer think that it can be published after a minor revision listed as follows.

1. The velocity field induced by the DBD actuators can be acquired in experiments using flow visualization method like particle image velocimetry (PIV) and the body force can also be derived from the velocity field measured. It is suggested to compare the velocity and body force field demonstrated in figure 6-8 with experimental results.

2. Could you please explain why the modified Suzen’s model is the one that fits the reference data better comparing with the modified shyy model and Dörr & Kloker model?

3. In figure 13, the results for the unmodified models is shown on the left side and the results obtained with the modified models is shown on the right. The explanation of this arrangement is suggested to be added in the figure title to improve readability.

4. In Case 1, the velocity and body force field acquired in simulation, as well as the force components is provided. In Case 2, the force component and velocity profile are provided. In Case 3, only the maximum induced velocity is demonstrated. What the reason for this discrepancy? Beside, the velocity field produced by both the Shyy and Dörr & Kloker models are demonstrated. Why the velocity field produced by the modified Suzen model is not demonstrated?

5. There is still a rather large discrepancy between the experiment and Suzen model, especially in the force component. How to further improve the perfoemance?

Author Response

We take your comments into account to improve the quality of the manuscript. We attach a file with the responses for each of the points.

Reviewer 2 Report

In this manuscript, three body force models of DBD plasma actuator were studied for evaluating their performance. Especially, some modifications were applied to these models for improving the capability reproducing the influence of operating parameters of DBD plasma actuator, and the effects of the modifications were investigated comparing with experimental results reported in some previous studies. The development of body force model is significant issue for promoting researches and practical applications of DBD plasma actuator, an if it were succeeded to develop a model which was able to quantitatively predict the body force of plasma actuator, it would have a great impact on the research community of plasma actuator. I agree with the importance of the study by authors, but unfortunately, the research output provided in the manuscript is still in progress and lack of completeness. Therefore, I cannot recommend the publication of the manuscript in current research stage. Specific reasons of my decision are as follows.

1)     In the manuscript, the performance of body force model was evaluated comparing with experimental results of some previous works by other research groups. But, the comparisons were inconsistent. In case 1, the comparison was performed from the viewpoint of body force estimated from flow velocity field, and on the other hand, in case 2, the body force from the method different from the method in case 1 was used. The y-profile of flow velocity was also compared, but the comparison was made using only maximum flow velocity in case 3. For accurate evaluation of the performance, the consistency of comparison with experiment is quite important. The several methods have been proposed for estimating body force from induced velocity field, however in my understanding, it is quite difficult to estimate the body force accurately due to assumptions in the method. I suggest authors to make comparisons using y-profile of flow velocity in all cases.

2)     The manuscript shows that the modifications of the models significantly improve their performance, however, the discrepancies from the experiment are still large, and further tuning of model parameters are required. For completeness of the study, authors had better to show a guideline to tune the model parameters.

3)    It is difficult to find what is original idea of authors, and how the original approach improve the model performance. The modification of the Suzen model and Dorr & Kloker model seems to come from the previous study by other research groups, and so the modified Shyy model is based on author’s original ideal. Authors had better to indicate the originality of the study more clearly and make discussions focusing on how the original approach works for improving the model performance.

Author Response

We appreciate your comments and review, we take them into account to improve the quality of the manuscript. The answers to each point can be found in the attached file.

In addition to the changes and corrections made under your comments, to improve the readability of the manuscript, a subtitle has been added to all subplots showing the results of the unmodified models on one side and the modified models on the other, indicating whether the results are for the unmodified or modified models. Figures 6-8 have also been modified to superimpose the force and shape of the wall jet (purple dotted line) and the force (black dotted lines) reported by Kotsonis et al.

Round 2

Reviewer 2 Report

Authors carefully responded to my questions and concerns, and the revisions authors made significantly improved the manuscript quality. However, I have still some concerns and conclude that some revisions are required before the publication to indicate the scope of this research more clearly. 

In my understanding, the body force estimated from flow velocity field includes large amount of estimation error due to the assumption adopted in the method. I have concerns about the quantitative comparison with the estimation results of body force. I consider that authors should mention the error included in the estimated body force in experiments and validity of the quantitative comparison.

Although the method proposed by authors significantly improved the capability of the model reproducing the body force strength, there are still discrepancies. Is the quantitative accuracy of the modified model proposed in the manuscript sufficient to reproduce the flow controllability of plasma actuator? Authors had better make some discussions that how much quantitative accuracy is required for the model to reproduce  the flow controllability. The discussions are quite valuable for readers who is considering to use the model. I recommend authors to check a following work which might be helpful for making the discussion.

Di Chen, et al., "A high-fidelity body-force modeling approach for plasma based flow control simulations," Physics of Fluids, 33, 037115, 2021.

Author Response

Thank you for your valuable feedback on our manuscript. We have carefully considered all of your comments and suggestions and have made the necessary changes to the manuscript.

A discussion of the uncertainty between the models and the reference force obtained from the velocity field has been added to the manuscript. It also comments on the validity of the models for their use in flow control studies.

We hope that these changes have addressed your concerns and that you find the revised manuscript satisfactory.

You will find our full responses in the attached file.

Round 3

Reviewer 2 Report

I concluded that the quality of the manuscript has reached the level for the publication by the current revisions.